# Laminin α4 Expression in Human Adipose Tissue Depots and Its Association with Obesity and Obesity Related Traits

**DOI:** 10.3390/biomedicines11102806

**Published:** 2023-10-17

**Authors:** Tobias Hagemann, Paul Czechowski, Adhideb Ghosh, Wenfei Sun, Hua Dong, Falko Noé, Christian Wolfrum, Matthias Blüher, Anne Hoffmann

**Affiliations:** 1Helmholtz Institute for Metabolic, Obesity and Vascular Research (HI-MAG) of the Helmholtz Zentrum München at the University of Leipzig and University Hospital Leipzig, 04103 Leipzig, Germany; 2Institute of Food, Nutrition and Health, ETH Zurich, 8093 Schwerzenbach, Switzerland; 3Medical Department III—Endocrinology, Nephrology, Rheumatology, University of Leipzig Medical Center, 04103 Leipzig, Germany

**Keywords:** *LAMA4*, laminin α4, obesity, RNA-seq, adipose tissue, metabolic health, type 2 diabetes

## Abstract

Laminin α4 (LAMA4) is one of the main structural adipocyte basement membrane (BM) components that is upregulated during adipogenesis and related to obesity in mice and humans. We conducted RNA-seq-based gene expression analysis of *LAMA4* in abdominal subcutaneous (SC) and visceral (VIS) adipose tissue (AT) depots across three human sub-cohorts of the Leipzig Obesity BioBank (LOBB) to explore the relationship between *LAMA4* expression and obesity (*N* = 1479) in the context of weight loss (*N* = 65) and metabolic health (*N* = 42). We found significant associations of *LAMA4* with body fat mass (*p* < 0.001) in VIS AT; higher expression in VIS AT compared to SC AT; and significant relation to metabolic health parameters e.g., body fat in VIS AT, waist (*p* = 0.009) and interleukin 6 (*p* = 0.002) in male VIS AT, and hemoglobin A1c (*p* = 0.008) in male SC AT. AT *LAMA4* expression was not significantly different between subjects with or without obesity, metabolically healthy versus unhealthy, and obesity before versus after short-term weight loss. Our results support significant associations between obesity related clinical parameters and elevated *LAMA4* expression in humans. Our work offers one of the first references for understanding the meaning of *LAMA4* expression specifically in relation to obesity based on large-scale RNA-seq data.

## 1. Introduction

Obesity and overweight remain to be serious challenges to global health, affecting ~60% of adults and one in three children in the European region alone [1]. Obesity is strongly associated with non-communicable diseases (NCD) [2] and thereby contributes to increased NCD mortality worldwide [3]. Thus far, overweight and obesity are risk factors for a variety of NCDs such as type 2 diabetes mellitus, coronary heart disease, and various forms of cancer [4,5,6]. In recent years, obesity has been associated with increased risk of mortality and severe disease causes in patients with COVID-19 [7] and long COVID-19 syndrome [8]. The mechanisms of how adipose tissue (AT) and adipocyte function may link increased fat accumulation in obesity to these diseases or adverse outcomes still needs to be explored.

The outer cell membrane of adipocytes is encapsulated by a basement membrane (BM) which provides cell stability [9] and is involved in cell signaling [10]. Aberrations of BM physiology are suggested to be related to pathological conditions such as disorganized BM structures in obese subjects [11] and stiffening of BM in association with diabetes [12]. One specific substructure of the BM is Laminin isoforms, which form heterotrimeric structures of α, β, and γ subunits [9]. The varying properties of Laminin isoforms allow for a variety of membrane structures and signaling [9], which change with tissue location and disease status [13]. The isoform laminin α4 (LAMA4) is known to be upregulated during adipogenesis [14] and has been proposed as a major molecule in the regulation of AT BM.

Analyses of the adipocyte surface proteome revealed an increased abundance of LAMA4 on the surface of adipocytes, both in diet-induced obesity (DIO) and in ob/ob mice [15]. Also, *LAMA4* seems to be expressed in a fat depot-dependent manner as LAMA4 abundance is significantly different between subcutaneous (SC) AT and visceral (VIS) AT in mice [15], in the human secretome [16], as well as in the human transcriptome [17]. Compared to wildtype mice, *LAMA4* −/− mice exhibit higher energy expenditure [13], decreased weight gain, and protection against diet-induced obesity and insulin resistance [18]. However, it is unclear whether these effects are due to *LAMA4*-loss induced expression change of laminin α5 [13,19]. Transcriptome analysis revealed that adipocyte-specific deletion of *LAMA4* seems to preserve the transcription patterns of healthy adipose tissue in DIO mice despite weight gain [20]. A more recent study reported an overall upregulation of *LAMA4* in SC white AT in both high-fat-diet (HFD)-fed mice and a human cohort of ten women with obesity, but without diabetes, compared to three healthy, lean women at the gene expression level [21]. In this study, SC AT *LAMA4* expression did not change with short term weight loss [21]. In addition to its potential relevance in obesity, *LAMA4* has been linked to the prognosis of gastric cancer and immune cell infiltration, suggesting *LAMA4* as a therapeutic target for these conditions [22].

Previous research has revealed a significant upregulation in *LAMA4* expression in obesity [21]. To further investigate the relationship between *LAMA4* and clinical markers of obesity, we conducted an analysis of *LAMA4* gene expression in SC and VIS AT depots across a total of 1586 individuals from three distinct clinical cohorts of the Leipzig Obesity BioBank (LOBB) using RNA sequencing data. We analyzed a large cross-sectional cohort comprising 1479 subjects, a cohort of 42 obese subjects classified based on their metabolic health (insulin resistance or sensitivity), and a cohort of 65 individuals with morbid obesity who underwent bariatric surgery. Our aim was to explore the relationship between *LAMA4* and clinical obesity markers, gender, weight reduction after bariatric surgery, and insulin resistance or sensitivity. Further, we investigated whether *LAMA4* expression appeared different among VIS or SC AT between different subgroups of individuals and tested the hypothesis that *LAMA4* expression decreases in both VIS and SC AT following bariatric surgery-induced weight loss. Finally, by combining Gene Set Enrichment Analysis (GSEA) with a comprehensive literature review, we identified statistically significant associations between *LAMA4* expression and obesity parameters.

## 2. Materials and Methods

### 2.1. Study Design of the Leipzig Obesity BioBank

The LOBB is a collection of human body fluids, adipose tissue samples, and associated data that was created to extend the knowledge of obesity and related diseases. The LOBB covers paired human samples of omental VIS and abdominal SC AT for three different cohorts. For the purpose of this study, samples have been collected in the period between 2008–2018 during elective laparoscopic abdominal surgery as previously described [23,24] and laboratory measurements of metabolic parameters and body composition were obtained, as detailed before [25,26]. Samples were collected under the following inclusion criteria: men and women at an age >18 years who underwent elective abdominal surgery and provided written informed consent before taking part in the study. Important exclusion criteria included: chronic drug or alcohol abuse, smoking during the last 12 months prior to surgery, acute inflammatory diseases, treatment with any medication affecting adipose tissue directly, end-stage malignant diseases, weight loss > 3% in the last 3 months prior to surgery, uncontrolled thyroid disease, and Cushing’s disease.

The cross-sectional cohort (CSC) covers 1479 individuals which were either normal/overweight (*N* = 31; 52% women; age: 55.8 ± 13.4 years old; BMI: 25.7 ± 2.7 kg/m^2^) or had obesity (*N* = 1448; 71% women; age: 46.9 ± 11.7 years old; BMI: 49.2 ± 8.3 kg/m^2^).

The two-step bariatric surgery cohort (BSC) was comprised of 65 individuals (66% women) with morbid obesity (BMI > 40 kg/m^2^) who all completed a two-step bariatric surgery approach with sleeve gastrectomy as a first step and ROUX-en-Y gastric bypass as a second step [23,24]. Patients included in this study had a preoperative BMI of 54.5 ± 9.3 kg/m^2^ and an age of 44.1 ± 9.2 years. After surgery, the patients average BMI was 40.9 ± 7.2 kg/m^2^ with an average age of 47.1 ± 9.9 years. On average, the patients lost 40.2 ± 21.2 kg between the two surgeries, and only individuals with a weight loss of more than five kilograms were included in the study. Preoperatively, type 2 diabetes (T2D) was diagnosed in 28 patients. The number of T2D cases dropped to 18 patients after the surgery. All individuals received frequent and structured healthy diet recommendations [27].

The cohort for distinguishing metabolically healthy versus unhealthy obesity (MHU) consisted of 42 individuals who were insulin resistant (IR; 71% female; age: 38.8 ± 11.1 years old; BMI: 45.9 ± 6.9 kg/m^2^; FPG: 5.2 ± 0.2 mmol/L; FPI: 27.9 ± 13.5 pmol/L) and 31 individuals who were insulin sensitive (IS; 71.43% female; age: 47.2 ± 7.7 years old; BMI: 47.3 ± 8.1 kg/m^2^; FPG: 5.7 ± 0.3 mmol/L; FPI: 113.7 ± 45.7 pmol/L) classified as previously described.

### 2.2. Bulk RNA-Sequencing and Analysis

Total RNA was extracted from SC and VIS AT and ribosomal RNA-depleted RNA sequencing data were generated following the SMARTseq protocol [28,29]. In brief, RNA was enriched and reverse-transcribed using Oligo(dT) and TSO primers. In silico PCR primers were employed for cDNA amplification and Tn5 was used to process cDNA with the Nextera DNA Flex kit (Illumina, San Diego, CA, USA). All libraries were sequenced as single-end reads on a Novaseq 6000 instrument (Illumina, San Diego, CA, USA) at the Functional Genomics Center Zurich, Switzerland.

Raw sequencing reads were preprocessed using *fastp* (v0.20.0) [30], with a minimum read length of 18 nts and a quality cut-off of 20. The pseudoaligner *kallisto* [31] was used for read alignment against the human reference genome (assembly GRCh38.p13, GENCODE release 32 [32]) and gene-level expression quantification. Samples with more than 20 million read counts were downsampled to 20 million read counts using the R package *ezRun* (v3.14.1; https://github.com/uzh/ezRun, accessed on 23 March 2022). Homoscedastic normalization with respect to library size was performed using the variance stabilizing transformation from *DESeq2* (v1.32.0) [33]. To avoid tissue variation in gene expression, we normalized each tissue separately as *LAMA4* expression was also significantly different between tissues.

To effectively mitigate the effects of in vitro RNA degradation, normalized counts were calibrated with respect to their transcript integrity numbers (TINs). TINs were estimated using the R package *RSeQC* (v4.0.0) [34]. Given the presence of a gender batch effect in the data, we performed an additional adjustment for gender when analyzing the combined dataset, irrespective of gender. However, when we stratified the data by gender, such an adjustment was not applied. Batch effects were adjusted using the *removeBatchEffect* function from *limma* (v3.56.2) [35] and prior examined using the R *swamp* package (v1.5.1) [36]. We did not detect an age-related batch effect, although there was a sometimes very large age difference between patients in the cohorts. Therefore, no adjustment for age was deemed necessary.

To avoid tissue variation in gene expression for correlation and gene set enrichment analyses, we normalized each tissue separately as *LAMA4* expression was also significantly different between tissues (Figure 1A). As there are variations in *LAMA4* expression observed between female and male individuals, and as we are specifically interested in examining gender-specific effects, we conducted separate normalization and batch correction for male and female subjects when performing correlation analyses.

Gene sets correlated with *LAMA4* were determined with genome-wide co-expression correlations using the *correlationAnalyzeR* (v1.0.0) package [37] to predict *LAMA4* gene function based on Gene Set Enrichment Analysis (GSEA) for Kyoto Encyclopedia of Genes and Genomes (KEGG) and Gene Ontology (GO) database annotations.

### 2.3. Statistical Analyses

Between-group comparisons were performed using a non-parametric statistical approach with the R package *ggstatsplot* v0.10.1 [38], based on one-way Kruskal–Wallis ANOVA and pairwise Mann–Whitney U test, corrected for multiple comparisons using the Hommel method. We performed univariate Spearman correlations between *LAMA4* expression and clinical phenotypes using the R package *RVAideMemoire* v0.9-81-2 [39], while accounting for multiple inferences by applying the false discovery rate (FDR) correction appropriate for the sample size. We considered absolute correlations ρ ≥ 0.1 and *p* < 0.5 as relevant. Analyses were conducted under R version 4.2.2 [40].

## 3. Results

### 3.1. Association of LAMA4 Gene Expression with Adiposity and Gene Set Pathway Alterations

In 1479 individuals of the cross-sectional cohort (CSC), we detected significantly higher *LAMA4* expression in VIS AT as in SC AT of both male and female individuals (all adj. *p* < 0.001; Figure 1A) as well as between individuals with and without obesity (adj. *p* < 0.001; Figure 1B). The difference in *LAMA4* expressed between the genders was less pronounced, but still existed. The median expression values of *LAMA4* in SC AT were 10.48 in females and 10.33 in males (adj. *p* = 0.01) and in VIS AT 10.91 in females and 11.09 in males (adj. *p* = 0.003, Figure 1A). Overall, no significant differences in *LAMA4* expression levels were found between individuals with and without obesity (Figure 1B).

Examining the CSC data for obesity-related signals associated with *LAMA4* expression we observed several correlations with a *p* < 0.05 and absolute Spearman’s correlation coefficient |ρ| ≥ 0.1 (not adjusted for FDR; Figure 1C, Appendix A). For instance, we observed associations of *LAMA4* with height (ρ = 0.14, *p* = 0.004, *N* = 426; Appendix A), HbA_1C_ (Hemoglobin A1c) (ρ = −0.17, *p* = 0.008, *N* = 230; Appendix A), and fasting plasma glucose (FPG; ρ = −0.13, *p* = 0.008, *N* = 393; Appendix A) in the SC AT of men. In the VIS AT of men, LAMA4 positively correlated with waist (ρ = 0.31, *p* = 0.009, *N* = 69; Appendix A) and interleukin 6 (IL-6; ρ = 0.66, *p* = 0.002, *N* = 19; Appendix A). In contrast, *LAMA4* positively correlated with body fat in VIS AT when gender was not taken into account (ρ = 0.13, *p* < 0.001, *N* = 669, Figure 1D), which was likely driven by the signal in women (ρ = 0.16, *p* < 0.001, *N* = 459, Figure 1E). After adjusting for multiple testing, only correlations with body fat in VIS AT (adj. *p* = 0.014) and the VIS AT of women (adj. *p* = 0.012) remained significant. Associations of *LAMA4* with BMI could not be found, neither in SC AT (ρ = −0.001, *p* = 0.96, *N* = 1479; Appendix A) nor in VIS AT (ρ = −0.01, *p* = 0.578, *N* = 1479; Appendix A). Due to missing values, *N* was alternating in the aforementioned correlation analysis.

The gene set enrichment analysis (GSEA, Appendix A) of gene sets correlated with *LAMA4* indicate that *LAMA4* is associated with the upregulation of pathways related neurodegenerative diseases such as Parkinson’s and Alzheimer’s, as well as respiratory diseases and ATP synthesis coupled electron transport pathways in both VIS (Figure 1F,G) and SC AT (Appendix A). In addition, we observed significant negative correlations between *LAMA4* and Uncoupling Protein 1 (*UCP1*), which served as a marker for thermogenic activity in both SAT (ρ = −0.52, *p* < 0.001, *N* = 1479) and VIS AT (ρ = −0.45, *p* < 0.001, *N* = 1479) (Appendix A). Conversely, maturity onset diabetes was found among the underrepresented pathways with regard to *LAMA4* expression in both fat depots (VIS AT: Figure 1H, SC AT: Appendix A).

In summary, we found significant correlation between VIS AT *LAMA4* expression and body fat, and significantly different expression levels between SC AT and VIS AT as well as between gender but not between obese- and non-obese individuals.

### 3.2. Extensive Weight Loss Does Not Strongly Affect AT LAMA4 Expression

We analyzed 65 individuals from the LOBB’s two-step bariatric surgery cohort (BSC) who underwent sleeve gastrectomy and gastric bypass intervention (time between surgeries: −2.94 ± 3.84 years, weight loss: 40.2 ± 21.2 kg) as described previously [23,24]. Consistent with the results in the CSC shown above, we found a significantly higher expression of VIS AT *LAMA4* compared to SC AT across both surgeries within the BSC (Figure 2A). However, within each tissue our analysis did not reveal any significant *LAMA4* expression changes between baseline and second step surgery.

When analyzing the BSC RNA sequencing data to identify obesity-related signals associated with *LAMA4* expression, we found several correlations that met the statistical criteria of *p* < 0.05 and |ρ| ≥ 0.1 (not adjusted for FDR; Figure 2B, Appendix A). Before surgery, we found negative correlations of *LAMA4* with body fat in VIS AT (ρ = −0.78, *p* = 0.005, *N* = 12) and SC AT (ρ = −0.78, *p* = 0.005, *N* = 12) obtained from men, but positive correlations with waist-to-hip ratio (WHR) in both SC (ρ = 0.45, *p* = 0.019, *N* = 25) and VIS AT (ρ = 0.38, *p* = 0.049, *N* = 25) when gender was not considered. Furthermore, pre-surgery *LAMA4* correlated positively with FPG (ρ = 0.31, *p* = 0.017, *N* = 56) and negatively correlated with hip circumference (ρ = −0.43, *p* = 0.024, *N* = 25). Post-surgery, we identified positive associations of *LAMA4* with IL-6 in SC AT (ρ = 0.87, *p* = 0.002, *N* = 8), C-reactive Protein (CrP) in male’s SC AT (ρ = 0.57, *p* = 0.005, *N* = 22) and serum Leptin in VIS AT (ρ = 0.84, *p* = 0.005, *N* = 8). At both time points, *LAMA4* expression correlated positively with WHR in SC AT (ρ = 0.33, *p* = 0.047, *N* = 33) and VIS AT (ρ = 0.34, *p* = 0.039, *N* = 33) (Figure 2B). No correlation remained significant after *p* adjustment.

In conclusion, we confirm higher *LAMA4* expression in VIS AT compared to SC AT and found correlations of *LAMA4* expression with WHR in both AT among others but did not find changes in *LAMA4* expression before and after surgery treatment.

### 3.3. AT LAMA4 Expression Is Not Related to Metabolically Healthy Obesity

To test the hypothesis that *LAMA4* expression is related to parameters of metabolic health, including preserved insulin sensitivity despite obesity, we performed group comparisons of AT *LAMA4* expression between IR and IS individuals [25,26] within the cohort, distinguishing metabolically healthy from unhealthy obesity (MHU). Our analysis did not reveal significant differences in *LAMA4* expression between IR and IS subjects between both fat depots (Appendix A). However, we observed a correlation between *LAMA4* expression and alanine transaminase (ALAT; ρ = −0.32, *p* = 0.039, *N* = 42) in IR SC AT, and positive correlations with HbA_1c_ of male IR individuals in SC AT (ρ = 0.68, *p* = 0.016, *N* = 12) and VIS AT (ρ = 0.60, *p* = 0.041, *N* = 12). In the SC AT of IS individuals, we found a negative correlation between *LAMA4* expression and mean adipocyte size (ρ = −0.41, *p* = 0.021, *N* = 31) as well as positive correlation with serum adiponectin (ρ = 0.46, *p* = 0.009, *N* = 31). After *p* adjustment, no correlation remained significant. Correlations are summarized in Appendix A and can be further explored in Appendix A.

In summary, we are not able to confirm differential *LAMA4* expression between IS and IR individuals but found *LAMA4* expression correlation with HbA_1c_ in males, among others.

## 4. Discussion

To the best of our knowledge, this study is one of the first investigations relating *LAMA4* expression to obesity-related health impairments with RNA sequencing technology. This *LAMA4* expression analysis of previously unprecedented scale provides a good overview of *LAMA4* interaction with clinical parameters.

In this study, we aimed at investigating the expression of *LAMA4* in SC and VIS AT depots across three distinct cohorts consisting of 1586 tissue donors from the LOBB. Specifically, we sought to explore the overall association between *LAMA4* expression and clinical parameters in a cross-sectional cohort primarily comprised of individuals with obesity. Moreover, we focused on investigating the potential differences in *LAMA4* expression between metabolically healthy and unhealthy individuals with obesity, as well as changes in *LAMA4* expression in response to extensive weight loss following bariatric surgery.

In both the cross-sectional and bariatric surgery cohorts, we found a significantly higher *LAMA4* expression in VIS AT compared to SC AT (Figure 1A,B and Figure 2A). This observation is comparable to similar results from a previous study by Goddi et al. [21]. While Goddi et al. report correlation between *LAMA4* in SC AT and body weight in HFD mice [21], we found significant associations between *LAMA4* and body fat in VIS AT (Figure 1C,D); WHR in both SC and VIS AT; and hip circumference in SC AT (Figure 2B). Also, it is worth noting that RNA sequencing data from Goddi et al. demonstrated higher expression of *LAMA4* in obese women when compared to lean women [21]. We were unable to confirm this finding in our cohort (Figure 1D), possibly due to our cohort’s structure comprising of individuals predominantly with obesity and a low number of individuals with normal or overweight. Nevertheless, our findings support previous reports [18,20,21] which associated *LAMA4* expression with obesity and increased fat mass.

Several studies have shown that *LAMA4* knock-out in mice resulted in healthier phenotypes and metabolisms [13,18,20]. Contrasting these results, we did not find associations of low *LAMA4* expression with metabolically healthy obesity. In accordance with previous results [21], we did not find changes in *LAMA4* expression after weight loss induced by bariatric surgery (Figure 2A). The absence of correlations between *LAMA4* and post-surgery WHR (Figure 2B) suggests that *LAMA4* expression is not affected by weight loss induced changes in adipose tissue function or cellular composition. Notably, we investigated whole AT biopsies and therefore cannot know whether or not *LAMA4* expression is altered at the adipocyte level.

Interestingly, gene enrichment analyses identified an overrepresentation of pathways relating *LAMA4* to mitochondrial activity and energy expenditure, (Figure 1F, Appendix A) supporting recent observations in *LAMA4* −/− mice [13]. Additionally, we found a strong correlation between *LAMA4* and *UCP1* (Appendix A), an indicator of thermogenic activity in (brown) adipocytes [41] suggesting that *LAMA4* is connected to energy expenditure signatures at the tissue level.

*LAMA4* knock-out in mice has been shown to preserve insulin sensitivity under DIO conditions [13], but we could not find significant expression differences between insulin-sensitive and insulin-resistant individuals in our MHU cohort (Appendix A). *LAMA4* was previously reported to be upregulated in adipogenesis [14] and might therefore alter AT function at early stages of AT formation, while changes in phenotype during adulthood might not affect *LAMA4* expression.

We need to acknowledge some limitations of our study. Although we did not include patients that reported smoking in the 12 months prior to surgery, we were not able to evaluate alcohol consumption as a potential confounding factor. Our cohort was composed of individuals predominantly with obesity and only a low number of individuals with normal or overweight could be included (*N* = 31). Despite the small sample size in the subgroups of metabolically healthy obesity with or without insulin resistance (*N* = 73) and in the 2-step surgery longitudinal cohorts (*N* = 65), these subgroups represent very well-characterized and, for the purpose of our study, relatively large groups. However, the lower patient numbers in these subgroups may be considered a limitation.

In conclusion, we show an association of *LAMA4* expression with parameters of fat accumulation and distribution in patients with obesity. However, at least in adults, *LAMA4* expression does not seem to be directly regulated by alterations in fat mass, obesity status, or metabolic health. Since we did not find differences in *LAMA4* expression between insulin-sensitive and insulin-resistant individuals with obesity, we propose that the previously suggested association of *LAMA4* with maintained metabolic health regardless of obesity requires altered *LAMA4* expression during early AT developmental stages. Our cohorts do not cover AT developmental stages, which is why we cannot comment on the developmental impact of *LAMA4* in AT within this study’s scope. Our data are indicative of associations between *LAMA4*, obesity, and adipose tissue function, but further studies are required to substantiate whether *LAMA4* is mechanistically linked to AT dysfunction and fat mass.

## Figures and Tables

**Figure 1 biomedicines-11-02806-f001:**
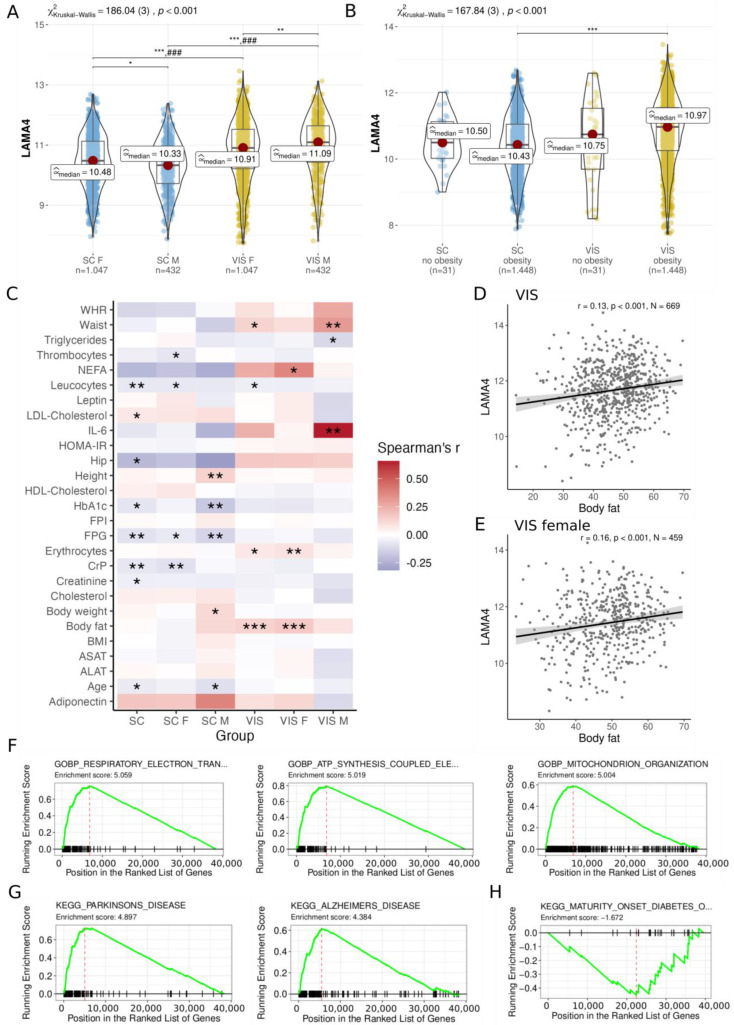
Analysis of *LAMA4* gene expression and correlation within the cross-sectional cohort. *LAMA4* expression comparison in SC and VIS AT of (**A**) male (M) and female (F); (**B**) individuals without obesity and obesity. Kruskal–Wallis one-way ANOVA and Dunn’s test for pairwise comparisons was conducted, which was corrected for multiple comparisons using the Hommel method. (**C**) *LAMA* gene expression correlation in SC and VIS AT of male and female individuals reveals several relationships with clinical parameters (unadjusted *p*-values). After *p*-value adjustment, *LAMA4* gene expression correlation remains significant with body fat in (**D**) VIS and (**E**) female VIS AT. Gene set enrichment analysis (GSEA, using the Gene Ontology database for biological processes) of the co-expressed genes with *LAMA4* in VIS AT reveals enrichment for pathways related to (**F**) metabolism and energy expenditure, (**G**) neurodegenerative diseases, and (**H**) a significant underrepresentation of the pathway for maturity onset diabetes. Significance levels are indicated as * < 0.05, ** < 0.01, *** < 0.001. *###* comparisons are significant in all groups of the other tissue (*p* < 0.001). ALAT: Alanine transaminase; ASAT: Aspartate transaminase; AT: adipose tissue; BMI: body mass index; CrP: C-reactive protein; F: female; FPG: fasting plasma glucose; FPI: fasting plasma insulin; HbA_1C_: hemoglobin A_1c_; HDL: High density lipoprotein; HOMA-IR: Homeostatic Model Assessment for Insulin Resistance; IL-6: interleukin 6; M: male; NEFA: nonesterified fatty acids; SC: subcutaneous; WHR: Waist hip ratio; VIS: visceral.

**Figure 2 biomedicines-11-02806-f002:**
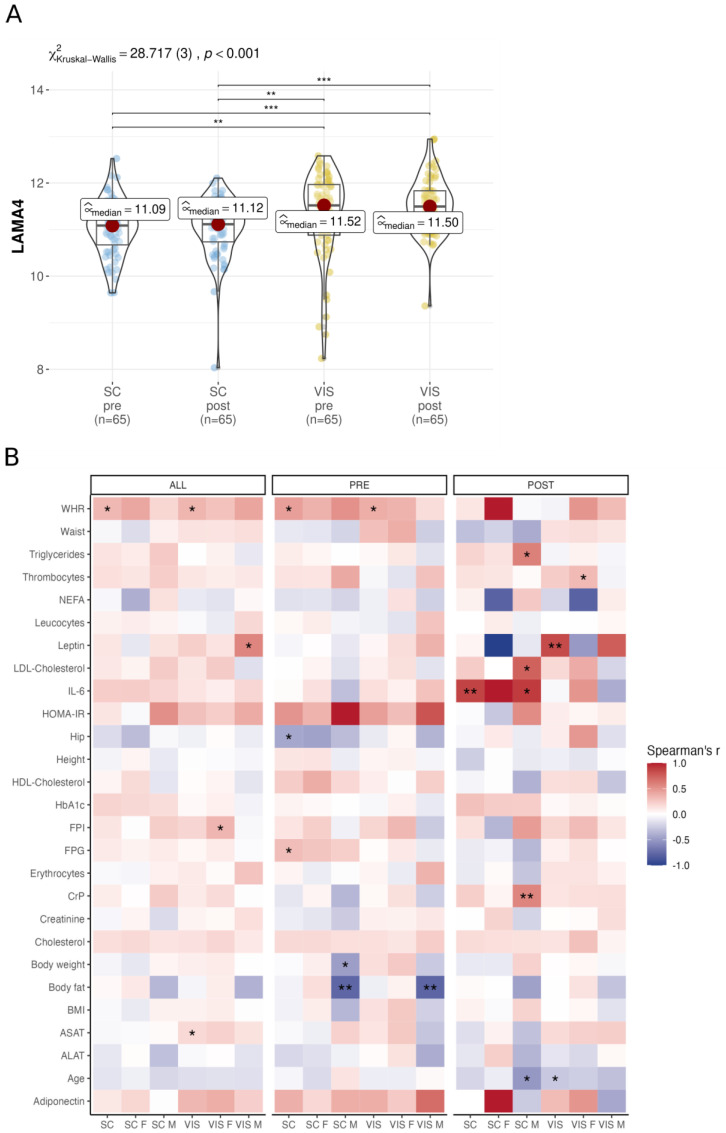
Analysis of LAMA4 gene expression and correlation within the bariatric surgery cohort. (**A**) Comparison of *LAMA4* expression pre- and post-surgery in SC AT and VIS AT, analyzed with Kruskal–Wallis one-way ANOVA and Dunn’s test for pairwise comparisons, corrected for multiple comparisons using the Hommel method. (**B**) Correlation analysis with Spearman correlation coefficient pre- and post-surgery and in both timepoints separated by gender and tissue (unadjusted *p*-values). Significance levels are indicated as * < 0.05, ** < 0.01, *** < 0.001 respectively.

## Data Availability

All data needed to evaluate the conclusions in the paper are present in the paper and/or the Appendix A. The human RNA sequencing data from the LOBB have not been deposited in a public repository due to restriction by patient consent but are available from the corresponding author on request.

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
