# Peer review of "Laminin α4 Expression in Human Adipose Tissue Depots and Its Association with Obesity and Obesity Related Traits"

_biomedicines, 2023, doi:10.3390/biomedicines11102806_

Round 1
Reviewer 1 Report
The paper entitled “ELAMA4 Expression in Human Adipose Tissue Depots and its 2 Association with Obesity, Metabolic Health, and Bariatric Surgery-induced Weight Loss” is interesting, but some major topics need improvements.
The study groups have several limitations that compromise the intrepretation of the resaults. The n of normal/overweight, weight loss and metabolic healthy groups are small. The differences in the age of the group participants are also relevant and not enable a corect comparison between groups.
The results must be simplified and focuses in the most relevant and significant results. The title of the paper is more ambitious than the conclusion achieved.
Reviewer 2 Report

Very minor edits needed
Reviewer 3 Report
See the attached review report
